# Cardiovascular Outcomes in the Acute Phase of COVID-19

**DOI:** 10.3390/ijms22084071

**Published:** 2021-04-15

**Authors:** Hiroki Nakano, Kazuki Shiina, Hirofumi Tomiyama

**Affiliations:** 1Department of Cardiology, Tokyo Medical University, Tokyo 160-0023, Japan; hiroki17@tokyo-med.ac.jp (H.N.); shiina@tokyo-med.ac.jp (K.S.); 2Department of Cardiology and Division of Pre-Emptive Medicine for Vascular Damage, Tokyo Medical University, 6-7-1 Nishishinjuku, Shinjuku-ku, Tokyo 160-0023, Japan

**Keywords:** COVID-19, arterial stiffness, cardiovascular disease

## Abstract

The cumulative number of cases in the current global coronavirus disease 19 (COVID-19) pandemic, caused by the novel severe acute respiratory syndrome coronavirus 2 (SARS-CoV-2), has exceeded 100 million, with the number of deaths caused by the infection having exceeded 2.5 million. Recent reports from most frontline researchers have revealed that SARS-CoV-2 can also cause fatal non-respiratory conditions, such as fatal cardiovascular events. One of the important mechanisms underlying the multiple organ damage that is now known to occur during the acute phase of SARS-CoV-2 infection is impairment of vascular function associated with inhibition of angiotensin-converting enzyme 2. To manage the risk of vascular dysfunction-related complications in patients with COVID-19, it would be pivotal to clearly elucidate the precise mechanisms by which SARS-CoV-2 infects endothelial cells to cause vascular dysfunction. In this review, we summarize the current state of knowledge about the mechanisms involved in the development of vascular dysfunction in the acute phase of COVID-19.

## 1. Introduction

SARS-CoV-2 is now recognized as causing not only fatal respiratory disease, but also fatal non-respiratory conditions, such as fatal cardiovascular events. Several factors, including conventional risk factors for cardiovascular disease (CVD), inflammation, oxidative stress, etc., are known to be involved in the pathogenesis of vascular damage [1]. The vasculature serves as the conduit for systemic circulation (i.e., the vasculature plays a role in efficient supply of blood to the peripheral organs), and both morphological and functional assessments of vascular damages are conducted in clinical practice. As for morphological assessment, imaging examinations such as computed tomography, magnetic resonance imaging and ultrasonography have been used to detect morphological vascular abnormalities such as stenosis, enlargement, torsion, plaque formation, aneurysm and dissection; on the other hand, functional vascular abnormalities are assessed by physiological or biochemical testing. Conventionally, the physiological indices used for the assessment of vascular functional abnormalities include the arterial stiffness, pressure wave reflection or endothelial function [2,3,4,5,6,7,8,9,10].

Conceivably, morphological vascular damage becomes apparent more gradually as compared to functional vascular damage. Therefore, functional assessment may be more sensitive than morphological assessment to assess vascular abnormalities occurring in the acute phase after exposure to factors causing vascular damage.

Poor cardiovascular outcomes have been recognized as being one of the serious problems that can occur during the acute phase of coronavirus disease 19 (COVID-19) [11,12,13]. Recent studies have demonstrated that increased arterial stiffness, a representative vascular functional abnormality, is an independent risk factor for cardiovascular diseases [14,15]. Arterial stiffness can arise from both structural abnormalities (degeneration of elastin fibers, vascular smooth cell hypertrophy and the proliferation of collagen fibers and/or vascular smooth muscle cells) and functional abnormalities (endothelial dysfunction, increased vascular tone and/or inner pressure) of the vasculature [5]. Viral infections can cause functional vascular damage [16,17,18], which could be initially reversible, via several mechanisms, such as through an increase in arterial stiffness. This functional increase in arterial stiffness may contribute to poor cardiovascular outcomes in COVID-19. In contrast to other viral infections, SARS-CoV-2 has been noted to inhibit angiotensin converting enzyme 2 (ACE-2) receptor activity [19,20,21,22]. The ACE2-Angiotensin 1-7 [Ang-(1-7)] axis is known to exert beneficial effects on vascular function, and such inhibition of the ACE-2 receptor activity may exacerbate the vascular dysfunction caused by viral infections by reducing the production of Ang-(1-7) [23,24,25,26,27]. In this review article, we discuss the clinical significance of the vascular dysfunction, especially functional increase in arterial stiffness, in the acute phase of COVID-19 in the context of its involvement in poor cardiovascular outcomes (Figure 1).

## 2. Basics of Vascular Function

The arterial wall consists of three layers: intima, media and adventitia. Each of the layers plays distinct roles in systemic circulation.

### 2.1. Functions of the Intima

The endothelium is the innermost layer of the arterial wall and comprises a single layer of endothelial cells. It contributes to the regulation of vasoconstriction/vasodilatation, vascular smooth muscle proliferation, hemostasis, inflammation and oxidation [28]. To maintain their homeostasis, endothelial cells synthesize several important vasoactive substances [29]. Among the most important of these is nitric oxide (NO), which plays an important role in protecting against atherosclerosis, acting on the vascular smooth muscle cells to cause vasodilatation, suppressing the activation of vascular smooth muscle cell proliferation and regulating thrombus formation and fibrinolysis via producing cell adhesion molecules, platelet aggregation and leukocyte adhesion to endothelial cells [28,29,30,31]. In addition, endothelial cells biosynthesize prostaglandin I_2_, which has a vasodilatory effect, and endothelin and angiotensin II, which have vasoconstrictive effects, all of which contribute to the regulation of the vascular tone.

### 2.2. Functions of the Media

The medial layer of the artery consists of circular smooth muscle cells and an extracellular matrix consisting of elastin fibers, collagen fibers and reticular fibers. The arterial tree, comprising a branching system of arteries, can be classified functionally into three types: elastic arteries, muscular arteries and arterioles. The most important role of the arterial tree is that it serves as the conduit that enables the constant supply of blood to the peripheral organs, and the medial layer contributes to maintaining the continuous blood flow via its cushioning effect. Especially in elastic arteries, the media has a high content of elastic fibers, and this elasticity converts the pulsating flow due to intermittent ejection of blood from the heart into a steady flow of blood to the peripheral organs (i.e., the Windkessel effect) [32]. This cushioning effect reduces the cardiac afterload and protects the arterial wall and microvascular system from the mechanical stress generated by cardiac contraction. Muscular arteries and arterioles are mainly composed of smooth muscle cells, and the former play a role in regulating the blood flow to organs via contraction/relaxation of the vascular smooth muscle, while constriction of the latter can increase the peripheral vascular resistance and affect the blood pressure. Vascular resistance in the arterioles is regulated via various mechanisms involving vasoactive substances and/or the autonomic nervous system; high blood pressure itself in the arterioles can stretch the smooth muscle cells in the medial layer and cause smooth muscles constriction, depending on the blood pressure [33,34].

### 2.3. Function of Adventitia

The adventitia consists of loose connective tissues, such as the extracellular matrix and fibroblasts, and plays a role in supporting the vascular architecture. It also contains blood vessels for nourishing the nerves and blood vessels that supply the walls of the blood vessels themselves (vasa vasorum). Vasa vasorum plays a role in supplying oxygen to layers of the blood vessels external to the media, where blood diffusion does not occur, and is found in blood vessels that are greater than 0.5 mm in thickness [35].

## 3. Pathophysiological Abnormalities Related to Endothelial Dysfunction/Arterial Stiffness

Common risk factors for arteriosclerosis, such as aging, obesity, hypertension, dyslipidemia, diabetes mellitus and smoking cause endothelial dysfunction by reducing NO production via inactivation of endothelial NO synthase (eNOS) and/or NO inactivation via enhanced production of reactive oxygen species (ROS) [36,37,38,39]. Endothelial dysfunction eventually also increases the structural arterial stiffness via increased vascular cell proliferation and collagen contents and/or vascular hypertrophy. Such increase in structural arterial stiffness contributes to poor cardiovascular outcomes [5,40].

On the other hand, cardiovascular events caused in the acute phase of COVID-19 infection may result, at least in part, from the shorter-term functional vascular dysfunction (Figure 2). Endothelial dysfunction caused by COVID-19 may increase platelet aggregation and the blood coagulation ability. The increased platelet aggregation and blood coagulation ability could potentially increase the risk of the development of thromboembolic events, such as acute coronary syndrome and stroke [41]. Endothelial dysfunction also causes inflammation of blood vessels and enhanced production of ROS. ROS reduce the activity of NO, increase the production of vasoactive substances, such as angiotensin II and endothelin I, and increase sympathetic nerve activity; the above act to cause contraction of the vascular smooth muscle cells in the media layer to increase functional arterial stiffness. Clinical studies have revealed the existence of significant associations of the brachial-ankle pulse wave velocity (baPWV), a marker of arterial stiffness, with markers of inflammation, oxidative stress, and a high sympathetic tone [42,43]. Increased functional arterial stiffness increases a summation of the forward pressure wave and reflected pressure wave in the aorta during cardiac systole via accelerating the speed of transmission of the pressure wave along the arterial wall [5,34,40]. This results in an increase of the cardiac afterload and in reduction of the stock volume of blood in the aorta, potentially contributing to impairment of the coronary arterial flow reserve and an increase in the risk of development of ischemic heart disease, myocardial disorders, arrhythmias and heart failure [44,45,46,47,48]. Attenuation of the cushioning effect of the elastic arteries increases the transfer of pulsatile energy to the peripheral microcirculation [40,49,50]. In high blood flow organs, such as the brain and kidneys, the pulsatile energy can penetrate through the microcirculation and cause microvascular damage.

## 4. Endothelial Dysfunction and Arterial Stiffness in Viral Infections

Viral infections cause endothelial dysfunction via several mechanisms, as follows: (1) the inflammatory response elicited by the infection directly activates the renin angiotensin system to increase the production of reactive oxygen species; (2) the viral infection can directly increase the production of reactive oxygen species and activate nuclear factor kappa B, which inactivates NO by converting it to peroxynitrite and reduces NO production by uncoupling it from endothelial NO synthase; (3) several cytokine receptors can directly or indirectly impair endothelial function. The endothelial dysfunction results in functional increase of the arterial stiffness. Viral infections may directly cause functional increase of the arterial stiffness by inducing the production of cytokines. Among these, interleukin-6 causes functional abnormality of the vascular smooth muscle cells, which could directly induce functional increase of the arterial stiffness [51]. Activation of neutrophils by viral infections can trigger various cellular mechanisms that can result in vasoconstriction, such as the release of prostanoids, lysosomal enzymes, as well as highly reactive oxygen species. Myeloperoxidases released from the azurophil granules of neutrophils play a particularly critical role in the attenuation of NO bioavailability [52]. These abnormalities also directly increase functional arterial stiffness.

## 5. ACE2/Ang-(1-7) Axis and Vascular Function

Ang-(1-7) has been demonstrated to protect against vascular endothelial damage via several mechanisms. In the initial step of vascular damage, endothelial cell adhesion molecules are activated and Ang-(1-7) suppresses the activation of these molecules. Ang-(1-7) reverses endothelial dysfunction via reducing the production of reactive oxygen species (ROS) [27], decreasing NADPH oxidase expression, and increasing endothelial NO synthase expression. Ang-(1-7) stimulates the proliferation of endothelial progenitor cells and also effectively prevents vascular smooth cell proliferation [53]. Furthermore, Ang-(1-7) counteracts hypercoagulability, which is known to cause vascular endothelial damage and is well-recognized to occur in the acute phase of COVID-19 [54]. 

## 6. Association of the ACE2/Ang-(1-7) Axis with the Vascular Function in Clinical Settings

Several studies have reported the association of the ACE2- Ang-(1-7) axis with endothelial function and/or arterial stiffness in clinical settings. Soro et al. reported that serum ACE2 activity correlated positively with the augmentation index, a marker of systemic arterial stiffness, in male type 1 diabetic patients [55]. This finding proposed that the ACE2-ang-(1-7) axis protects against the development of diabetic vasculopathy, which increases the arterial stiffness. Durand et al. demonstrated that Ang-(1-7) treatment restores the NO component of flow-mediated vasodilatation (FMD), a marker of endothelial function, in arterioles isolated from patients with coronary artery disease [56]. Srivastava et al. demonstrated an association between the serum levels of Ang-(1-7) and the FMD and PWV in patients with diabetes mellitus [57]. Furthermore, they also reported that in patients with diabetes mellitus, an angiotensin-converting enzyme inhibitor treatment for 3 months improved the FMD and reduced the carotid-femoral pulse wave velocity, a marker of the arterial stiffness [58].

## 7. Increased Arterial Stiffness-Related Cardiovascular Morbidities in COVID-19

Limited data are available on the association of COVID-19 with vascular function. Ratchford et al. reported observing endothelial dysfunction and increased arterial stiffness in young adults with COVID-19 [59]. Thus, COVID-19 may cause endothelial dysfunction and increased arterial stiffness. As mentioned above, these may cause functional cardiovascular abnormalities, such as an increase of the cardiac afterload, impairment of the coronary arterial blood supply, and/or microvascular damage. Furthermore, endothelial dysfunction may also cause hypercoagulability, and increased arterial stiffness may also cause endothelial dysfunction via increasing the tensile and shear stress on the vascular endothelium [60,61,62]. These abnormalities could contribute to the development of heart failure, arrhythmias, coronary ischemia, acute coronary syndrome and stroke.

## 8. COVID-19 and Cardiovascular Morbidities

### 8.1. COVID-19 and Cardiovascular Morbidities

A number of reports have been published on the association between COVID-19 and CVD. Hypertension (30%), diabetes (19%) and coronary artery disease (8%) are the most commonly reported comorbidities in patients with COVID-19 [63,64], and these comorbidities are associated with an increased risk of aggravation of the cardiovascular outcomes in COVID-19 [65,66,67]. The elevated risk of development of new cardiovascular events is observed not only in COVID-19 patients with a previous history of CVD, but also in those without a previous history of CVD [68,69,70]. Numerous clinical studies have reported elevated serum troponin I levels, a marker of myocardial injury, in patients with COVID-19. The prevalence rate of such an elevation was 7–17% of patients in a general cohort and 36% in a hospitalized cohort [63,71,72,73,74]. Patients with a history of hypertension (60%) and coronary artery disease (30%), in particular, have high incidences of an elevation of troponin I levels [41]. Higher levels have also been reported to be predictive of a higher mortality rate [75].

Although it is possible that non-ischemic myocarditis is what underlies the myocardial injury in COVID-19, there are currently no data to indicate the presence of SARS-Cov-2 virus in myocardial tissues. On the other hand, attenuation of the ACE-2 receptor activity in COVID-19 may cause an increase of the cardiac afterload via elevation of the blood pressure and/or increase of the arterial stiffness, impairment of myocardial microcirculation and decrease of the coronary arterial flow reserve. These pathophysiological abnormalities, along with myocardial injury due to hypoxemia and severe respiratory failure are considered to contribute to elevated mortality rates in patients with COVID-19.

### 8.2. COVID-19 and Acute Coronary Syndrome

In patients with COVID-19, vascular endothelial damage due to an acute inflammatory reaction may trigger platelet aggregation and blood coagulation, and also increase the production of cytokines, such as interleukin (IL)-2, IL-7, IL10, granulocyte-colony stimulating factor, monocyte chemoattractant protein-1 and tumor necrosis factor-α [73,76]. These abnormalities may increase the risk of myocardial infarction caused by plaque rupture or thromboembolism. Although acute viral respiratory infections have been reported to be associated with an increased risk of occurrence of acute coronary syndrome [77], it has not yet been clarified whether COVID-19 could also be a causal factor for acute coronary syndrome. Stefanini et al. reported 28 cases of COVID-19 with ST segment elevation on the electrocardiogram. Among these patients, coronary angiography revealed a culprit lesion requiring revascularization in 60.7% [78]. On the other hand, no evidence of any obstructive coronary artery disease was noted in the remaining 39.3% of patients, suggesting the involvement of type 2 myocardial infarction associated with hypoxemia due to lung injury. Furthermore, Fizzah et al. reported that 33.9% of 115 consecutive patients with ST elevation-myocardial infarction (STEMI) diagnosed by coronary angiography showed positive test results for COVID-19 [79]. The serum levels of high-sensitivity troponin, D-dimer, and C-reactive protein were higher in the STEMI patients with COVID-19 than in those without COVID-19, and coronary multivessel thrombosis and stent thrombosis, indicative of hypercoagulability, were frequently observed in the STEMI patients with COVID-19. Data needed to assess the association of COVID-19 with the incidence of acute coronary syndrome are still limited. However, attention needs to be paid to the possible occurrence of acute coronary syndrome in patients with COVID-19.

### 8.3. COVID-19 and Arrhythmic Diseases

In one observational study, tachyarrhythmias and bradyarrhythmias were observed in 16.7% of hospitalized patients with COVID-19, and more frequently in critically ill patients admitted to the intensive care unit (44.4%) [74]. In addition, there have been reports of critical events leading to cardiac arrest during hospitalization, even in COVID-19 patients without a previous history of CVD, but the details of these arrhythmias are unclear [80]. Arrhythmias associated with COVID-19 have been considered to be caused by myocardial damage, inflammation, repolarization of myocardial conduction due to the action of inflammatory cytokines on the myocardial ion channels and some drugs used for COVID-19 treatment [81]. On the other hand, multiple complications are known to occur in patients admitted to the intensive care unit (ICU), and 15.7% and 19.7% of patients of the surgical ICU and the medical ICU developed arrhythmias, respectively [82]. It is worthy of note that according to one observational study, 78% of critically ill patients admitted to the ICU had arrhythmias [83]. Not only myocardial damage, but also hypoxemia due to viral respiratory infection, systemic inflammation, metabolic disorders, activation of neurohormonal factors and other patient backgrounds may affect the risk of the development of arrhythmias. Furthermore, increased cardiac afterload associated with increased arterial stiffness may, at least in part, affect the arrhythmogenic state. Further studies are needed to delineate the factors associated with arrhythmias in patients with COVID-19.

### 8.4. COVID-19, Cardiomyopathy and Myocarditis

Patients with COVID-19 have acute myocardial injury in the absence of coronary artery disease [84]. In the case series reported by Arentz et al., 33% of COVID-19 patients developed cardiomyopathy, although the mechanism remained unclear [85]. There are reports of takotsubo cardiomyopathy or sepsis-induced cardiomyopathy due to catecholamine toxicity and cytokine storms, which is reversible cardiac dysfunction, in critically ill patients with COVID-19 [86]. Myocarditis, which causes heart failure, has been reported in many patients with COVID-19. However, the direct association of SARS-CoV-2 infection with the development of myocarditis has not yet been clarified. Several mechanisms of inducing cardiomyopathy and myocarditis have been hypothesized. The first, one of the key biochemical pathways involved in cardiomyopathy and myocarditis is NOD-, LRR- and pyrin domain-containing protein 3 (NLPR3) inflammasome which increased production of pro-inflammatory and atherogenic cytokines inducing multi-organ failure [87]. The second, imbalanced T helper 1 and T helper 2 responses related to cytokine release. This could lead to apoptosis and necrosis of cardiomyocytes [41,88].

### 8.5. COVID-19 and Heart Failure

The incidence of heart failure in hospitalized patients with COVID-19 has been reported to be 23.0%, and heart failure is one of common causes of in-hospital death [89]. Although it is difficult to completely distinguish acute heart failure from acute respiratory failure, it was reported that heart failure was responsible for 52% of the deaths from COVID-19 in Wuhan [63]. According to another study, the mortality rate of hospitalized COVID-19 patients with a history of heart failure was 24.2%, and age, male sex, and obesity were identified as risk factors for mortality [90]. Moreover, the mortality of heart transplant recipients was 33.3%, and right ventricular dysfunction, arrhythmias, thromboembolic events and markedly elevated cardiac biomarkers were indicated as risk factors of mortality [91]. Gao et al. showed that the serum levels of NT-proBNP, which is one of the markers of heart failure, were correlated with in-hospital mortality in patients with COVID-19, and were also significantly associated with the serum levels of systemic inflammation and elevated myocardial injury markers, suggesting that this parameter could be useful as one of the indicators of the severity of COVID-19 [92]. A possible mechanism of the development of heart failure is that an increase of the arterial stiffness via vascular endothelial damage, directly or via ACE2, causes an increase of the cardiac afterload and microcirculatory disorder [64]. In addition, acute lung injury, including acute respiratory distress syndrome, causes pulmonary hypertension and right heart failure. It has been reported that rapidly progressive cardiac dysfunction, such as sepsis-related cardiomyopathy and takotsubo cardiomyopathy, may also be among the causes of heart failure in patients with COVID-19.

## 9. Cardio-Oncology and Cardiovascular Outcomes in COVID-19

Cancer patients are vulnerable with a high risk of viral infection and history of some anti-cancer therapies, and are expected to be at higher risk of developing cardiovascular complications with COVID-19, such as cardiotoxic chemotherapy, radiotherapy and immune checkpoint inhibitors. Several recent retrospective studies showed that cancer patients had a higher risk of SARS-CoV-2 infection and had a higher probability of severe events than non-cancer patients with COVID-19 [93,94]. Guen et al. reported that cancer patients had high risk for admission to the ICU, invasive ventilation and death compared to non-cancer patients [95]. In especially, a retrospective cohort study on 28 cancer patients with COVID-19 indicated that recent surgery, chemotherapy and radiotherapy had risk of SARS-CoV-2 infection [96]. Systematic reviews showed that the prevalence of cancer patients with COVID-19 was 0.92–2%, and there was 7.6% higher mortality than in non-cancer patients [97,98]. Cardiovascular vulnerability is thought to one of the causes of poor outcome in cancer patients with COVID-19 and several mechanisms have been indicated [99]. First, several risk factors such as hypertension, aging and obesity exacerbate both COVID-19 and cancer treatment related cardiovascular complications. Second, the patients with cardiotoxic chemotherapy also have reduced cardiac reserve and might be severe cases of COVID-19. Moreover, cardiotoxic cancer treatment causes cardiovascular damage due to oxidative stress, pro-apoptotic and inflammation, and enhances the cardiovascular event in patients with COVID-19. Cardiotoxic drugs include anthracyclines, human epidermal growth factor receptor 2 (HER2) blocking antibodies, tyrosine kinase inhibitors, proteasome inhibitors and immune checkpoint inhibitors. These drugs cause apoptosis, oxidative stress and necrosis in cardiomyocyte and increase the risk of heart failure, cardiomyopathy and fatal myocarditis [88,100,101,102,103]. It is speculated that cancer patients with cardiotoxic chemotherapy had high risk of incidence of cardiovascular event. Finally, increased arterial stiffness in acute phase COVID-19 may exacerbate their cardiovascular outcomes. Further studies are needed to clarify the association of cancer patients and COVID-19.

## 10. Arterial Stiffness and Cardiovascular Outcomes in COVID-19

Rodilla et al. attempted to estimate the arterial stiffness, reflected by a wide pulse pressure, in COVID-19 patients requiring hospitalization and analyze its association with all-cause in-hospital mortality [104]. They analyzed the data of an observational, retrospective, multicenter cohort study conducted in 12,170 patients admitted to 150 Spanish centers included in the Spanish Society of Internal Medicine (SEMI)-COVID-19 Network, to examine the association of the arterial stiffness with all-cause in-hospital mortality. Overall, 2606 (21.4%) subjects died during the first 50 days of follow-up. Multivariate analysis identified elevated arterial stiffness and a systolic blood pressure of <120 mm Hg as being significant and independent predictors of the all-cause in-hospital mortality. The authors described that that age and elevated blood pressure affected the increased arterial stiffness, but they did not examine the evolutional changes of arterial stiffness during the hospital admission. Even so, the findings of their study suggest that increased arterial stiffness contributes to poor cardiovascular outcomes in the acute phase of COVID-19.

## 11. Conclusions

COVID-19 may cause vascular function abnormalities, as reflected by an increase in the arterial stiffness. Because of the inhibition of ACE-2 receptor activity in COVID-19, the vascular function abnormalities may be exaggerated as compared to those in infections caused by viruses other than SARS-CoV-2. This may contribute, at least in part, to the poor cardiovascular outcomes in the acute phase of COVID-19. The changes in the functional arterial stiffness during the clinical course of COVID-19 could be assessed by measurement of the pulse wave velocity (PWV). Further studies are needed to examine whether the PWV might be useful as a marker for monitoring the cardiovascular risk status in the acute phase of COVID-19.

## Figures and Tables

**Figure 1 ijms-22-04071-f001:**
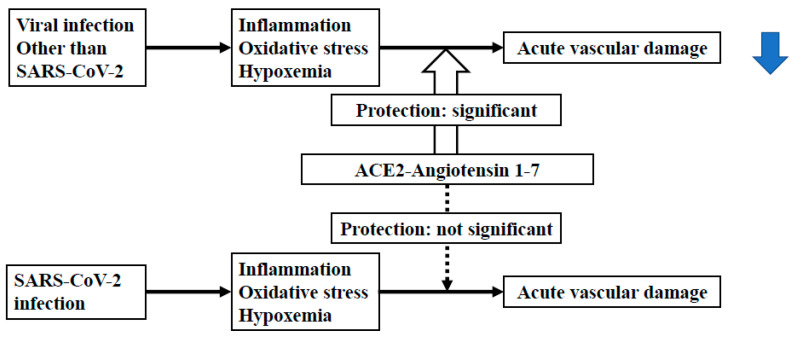
Comparison of the effects of severe acute respiratory syndrome coronavirus 2 (SARS-CoV-2) infection and other infections on acute vascular dysfunction via angiotensin converting enzyme 2 (ACE2)-Angiotensin 1-7.

**Figure 2 ijms-22-04071-f002:**
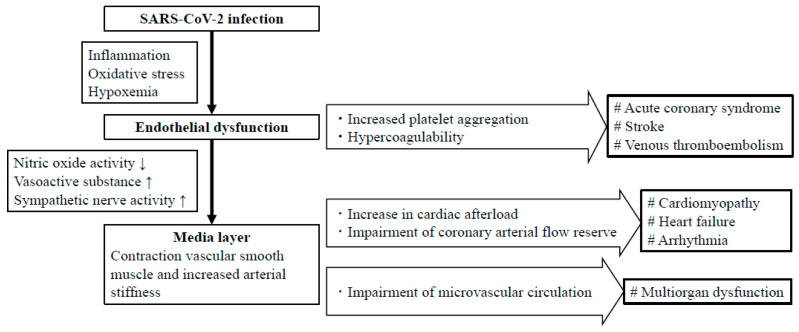
Mechanisms of cardiovascular dysfunction in acute phase of COVID-19.

## Data Availability

Not applicable.

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
