# Peer review of "Cardiovascular Outcomes in the Acute Phase of COVID-19"

_ijms, 2021, doi:10.3390/ijms22084071_

Round 1
Reviewer 1 Report
The manuscript titled " Cardiovascular Outcomes in the Acute Phase of COVID-19" is a high quality review describing the cardiovascular outcomes in patients with COVID-19. The manuscript is well written and references are of high quality. However, we suggest to add more information about the clinical effects of COVID-19 in cardio-oncology patients ( you can cite https://doi.org/10.3390/cancers12113316 ). The authors should describe in detail the outcomes seen in patients with cancer, with particular attention of cancer survivors and the vulnerability of patients treated with high toxic anticancer therapies like doxorubicin (you can cite doi: 10.2459/JCM.0000000000000378). Moreover, authors should describe in a better way the key biochemical pathways involved in myocarditis, VTE and heart failure associated to COVID-19 infection ( as example, the Inflammasome complex , see doi: 10.26355/eurrev_202009_22867 ). The manuscript will be acceptable after minor revision.
Reviewer 2 Report
The review work presented by Hiroki Nakano and et al. is well written, clear, and easy to read. It adds clustered information to the subject area in the Acute Phase of COVID-19 for what concern cardiovascular outcomes.
I noticed that amongst the cardiovascular situation described by authors the cardiac transplantation is not present (please see https://doi.org/10.3390/reports3030023), might can be useful to add this case report as well in the 8.5 section.
Minor:
Please do not leave space in the abstract.
Please correct the 6th paragraph as the others, also the format writing style.
